

# A preliminary case study of the effect of shoe-wearing on the biomechanics of a horse's foot

Olga Panagiotopoulou[1,2], Jeffery W. Rankin[1], Stephen M. Gatesy[3] and John R. Hutchinson[1,2]

[1] Structure & Motion Laboratory, Department of Comparative Biomedical Sciences, The Royal Veterinary College, Hatfield, Hertfordshire, United Kingdom
[2] Moving Morphology & Functional Mechanics Laboratory, School of Biomedical Sciences, Faculty of Medicine and Biomedical Sciences, The University of Queensland, Brisbane, Australia
[3] Department of Ecology and Evolutionary Biology, Brown University, Providence, RI, USA

Corresponding authors
Olga Panagiotopoulou,
o.panagiotopoulou@uq.edu.au
John R. Hutchinson,
jhutchinson@rvc.ac.uk,
jrhutch@rvc.ac.uk

## ABSTRACT

Horse racing is a multi-billion-dollar industry that has raised welfare concerns due to injured and euthanized animals. Whilst the cause of musculoskeletal injuries that lead to horse morbidity and mortality is multifactorial, pre-existing pathologies, increased speeds and substrate of the racecourse are likely contributors to foot disease. Horse hooves have the ability to naturally deform during locomotion and dissipate locomotor stresses, yet farriery approaches are utilised to increase performance and protect hooves from wear. Previous studies have assessed the effect of different shoe designs on locomotor performance; however, no biomechanical study has hitherto measured the effect of horseshoes on the stresses of the foot skeleton *in vivo*. This preliminary study introduces a novel methodology combining three-dimensional data from biplanar radiography with inverse dynamics methods and finite element analysis (FEA) to evaluate the effect of a stainless steel shoe on the function of a Thoroughbred horse's forefoot during walking. Our preliminary results suggest that the stainless steel shoe shifts craniocaudal, mediolateral and vertical GRFs at mid-stance. We document a similar pattern of flexion-extension in the PIP (pastern) and DIP (coffin) joints between the unshod and shod conditions, with slight variation in rotation angles throughout the stance phase. For both conditions, the PIP and DIP joints begin in a flexed posture and extend over the entire stance phase. At mid-stance, small differences in joint angle are observed in the PIP joint, with the shod condition being more extended than the unshod horse, whereas the DIP joint is extended more in the unshod than the shod condition. We also document that the DIP joint extends more than the PIP after mid-stance and until the end of the stance in both conditions. Our FEA analysis, conducted solely on the bones, shows increased von Mises and Maximum principal stresses on the forefoot phalanges in the shod condition at mid-stance, consistent with the tentative conclusion that a steel shoe might increase mechanical loading. However, because of our limited sample size none of these apparent differences have been tested for statistical significance. Our preliminary study illustrates how the shoe may influence the dynamics and mechanics of a Thoroughbred horse's forefoot during slow walking, but more research is needed to quantify the effect of the shoe on the equine forefoot during the whole stance phase, at faster speeds/gaits and with more individuals as well as with a similar focus on the hind feet. We anticipate that our preliminary analysis using advanced methodological approaches will pave the way for new directions in research

on the form/function relationship of the equine foot, with the ultimate goal to minimise foot injuries and improve animal health and welfare.

## INTRODUCTION

Horse racing is a multi-billion-dollar, worldwide industry in which the welfare of the horses is of paramount importance. Musculoskeletal injuries are both a common cause of economic loss within the industry and a major welfare concern due to the resulting morbidity and mortality (*McKee*, *1995*; *Jeffcott et al.*, *1982*; *Clegg*, *2011*; *Bailey et al.*, *1999*). The cause of musculoskeletal injuries is multifactorial: pre-existing pathologies, increased speeds, and track surfaces are all recognised as contributing factors (*Parkin et al.*, *2004*; *Cogger et al.*, *2006*; *Foote et al.*, *2011*; *Clegg*, *2011*). Horses have evolved to only maintain their third digit, which ends in a rigid hoof capsule and is functionally adapted to fast speeds (*Dyce, Sack & Wensing*, *2010*). The hoof and the interphalangeal joints receive most of the impact loads when the foot hits the ground (*Dyhre-Poulsen et al.*, *1994*) and at fast speeds these loads can exceed 2.5 times the horse's body weight (*Witte, Knill & Wilson*, *2004*). Under load-bearing conditions, the distal and coronary borders of the hooves expand (*Colles*, *1989*); the dorsal hoof wall rotates caudoventrally about the third digit and the heel expands between 2–4 mm (*Jordan et al.*, *2001*).

Farriery (horseshoe design) approaches in both domestic and racehorses have been used since the domestication of horses to protect hooves from wear and to allow manipulation of the shape of the foot to improve performance and enhance biomechanical function. Nevertheless, different horseshoe materials have varying effects on horses' feet due to their wide range of weight, toe angle, frictional and damping properties and their interaction with foot trimming (*Roepstorff, Johnston & Drevemo*, *1999*; *Pardoe et al.*, *2001*; *Van Heel et al.*, *2005*; *Van Heel, Van Weeren & Back*, *2006*; *Heidt et al.*, *1996*; *Willemen, Savelberg & Barneveld*, *1998*). Previous *in vivo* studies in horses have shown that an elevation of the hoof due to the presence of the shoe increases the pressure within the distal interphalangeal joint, which may account for an increase of bone stresses that can enhance the development of degenerative joint diseases (*Roepstorff, Johnston & Drevemo*, *1999*). Although no biomechanical study to date has quantified bone stresses of the horse forefoot in the shod and unshod conditions *in vivo*, *Moyer & Anderson* (*1975*) hypothesised that increased loading due to farriery can increase stresses on the horse foot and lead to injuries (*Moyer & Anderson*, *1975*). In addition, an *ex vivo* analysis by *Ault et al.* (*2015*) recorded significant increases in the strain of the superficial digital flexor tendon (SDFT) and the suspensory ligament of shod horses, further supporting the inference that shoes disrupt the natural ability of horses' feet to maintain tendon (and perhaps other tissue) strains at lower levels.

Despite the likelihood that shoes impact equine digit function, current knowledge of the relationships between foot function, farriery approaches and musculoskeletal injury is

limited, partly due to the lack of an established *in vivo* experimental protocol for studying foot dynamics and mechanics. Finite element analysis (FEA) is a numerical technique well entrenched in equine biomechanics as a tool to measure deformation (stress, strain) in complex continuous systems (such as the hoof), by dividing them into sub-regions of finite size (elements) using linear ordinary differential equations (*Hutton, 2003*). With FEA, scientists have managed to study the deformations of anatomically deep structures of the equine distal foot in shod and unshod conditions (*Harrison et al., 2014*; *Hinterhofer, Stanek & Haider, 2001*; *Hinterhofer, Stanek & Binder, 1998*; *Bowker et al., 2001*; *Salo, Runciman & Thomason, 2009*; *O'Hare et al., 2013*; *Thomason, Douglas & Sears, 2001*; *Thomason, McClinchey & Jofriet, 2002*; *Thomason et al., 2005*; *McClinchey, Thomason & Jofriet, 2003*; *Collins et al., 2009*; *Douglas et al., 1998*). These studies have enhanced our understanding of how the equine digit deforms under load-bearing, but more robust *in vivo* data and subject-specific models are needed to fully characterize how the equine distal limb's functional environment relates to disease. This requires combining accurate joint motion and ground reaction force (GRF) data from a synchronised time sequence with subject-specific bone geometry. Here we show how a combination of different techniques can be used to obtain these data to generate high fidelity *in vivo* FEA results.

A common approach for researchers to measure joint motion in horses is the attachment of motion analysis markers on the skin overlying bony structures. This approach introduces errors, due to artefacts from skin and hoof motion, which can be as large as the actual joint motion (*Reinschmidt et al., 1997*; *Roach et al., 2015*). One alternative to surface skin markers is the surgical implantation of intra-cortical bone pins into the limb bones, but this methodology is highly invasive (e.g., *Clayton et al., 2004*; *Clayton et al., 2007a*; *Van Weeren, Van den Bogert & Barneveld, 1990*; *Chateau, Degueurce & Denoix, 2004*). Although these pins can more accurately quantify bone motion, their invasiveness may affect the natural function/behaviour of the joints (*Lundberg et al., 1989*) and are inappropriate to use when studies require a large number of horse participants. Fortunately, a new alternative technology using biplanar radiography, commonly referred to as X-ray Reconstruction of Moving Morphology or XROMM, has been developed that can be used to accurately characterize the three-dimensional (3D) motion of joints (*Brainerd et al., 2010*; *Gatesy et al., 2010*).

XROMM combines bi-planar fluoroscopic images to track dynamic functions such as trotting, which enables precise measurements of joint motion without artefacts from soft tissue motion (e.g., *Miranda et al., 2013*). By acquisition of fluoroscopic images in two planes and with the assistance of specialised software, the images can be combined to track motion of individual skeletal elements in three dimensions. Thus, motion can be assessed *in vivo*, without the requirement for attachment of any device to the skin or into the bones. Natural behaviour can be measured in a manner not possible with other techniques and with minimal risk to the animal/participant, while keeping radiation doses reasonably low. To date, the XROMM marker-based (*Astley & Roberts, 2012*; *Baier & Gatesy, 2013*; *Brainerd, Moritz & Ritter, 2016*; *Brainerd et al., 2010*; *Camp, Roberts & Brainerd, 2015*; *Camp & Brainerd, 2014*; *Dawson et al., 2011*; *Gidmark et al., 2013*; *Heers et al., 2016*; *Kambic, Roberts & Gatesy, 2014*; *Kambic, Roberts & Gatesy, 2015*; *Nowroozi & Brainerd, 2013*) and markerless (*Baier, Gatesy & Dial, 2013*; *Falkingham & Gatesy, 2014*;

*Gatesy et al.*, *2010*; *Nyakature & Fischer*, *2010*) technology has been used to study diverse behaviours such as the limb kinematics of frogs, birds, bats and dogs; jaw kinematics during feeding in fish, pigs, birds and bats and rib kinematics of breathing in lizards.

This study presents a novel method that combines three-dimensional data from XROMM (*Brainerd et al.*, *2010*; *Gatesy et al.*, *2010*), inverse dynamics methods, and finite element analysis to perform a preliminary investigation of the effect of a stainless steel shoe on the function of one Thoroughbred horse's foot during walking. The intent of this work is not to draw clinical conclusions on the effect of the shoe on equine foot mechanics. Instead, we present a methodological approach that can be used in future research to study the effect of different shoe designs on foot mechanics and potentially inform the design of new shoes that can improve locomotor performance while maintaining the integrity of musculoskeletal structures.

## MATERIALS & METHODS

### Subjects

One Thoroughbred healthy male adult horse (540 kg body mass) from the Royal Veterinary College (RVC) participated in the study. The horse had previously been trained for and participated in locomotor studies in the laboratory. Fifteen minutes of training were provided for the horse to adapt to the experimental setup. The study was reviewed and approved by the Royal Veterinary College's Ethics and Welfare Committee (approval number URN 2011 1094).

### Data collection

Each trial lasted two to four seconds, during which the horse was led across a custom-designed platform (Fig. 1A). A custom-designed platform rather than a treadmill was used for this study because our methodological approach for the *in vivo* estimation of the intersegmental forces for FEA required accurate ground reaction force (GRF) measurements. Accurate measurements of all GRF components could not be obtained using any available treadmill.

A Sony HDR (Sony, London, UK) high definition video camera was placed perpendicular to the platform to approximate walking speed (25 Hz). To obtain foot kinematics, two custom X-ray fluoroscopes (RSTechnics, Netherlands; refurbished Phillips systems, 36 cm intensifier; ≤110 kV, ≤3 mA) were retrofitted with two AOS high-speed digital cameras (AOS Technologies AG, Switzerland) to acquire biplanar fluoroscopy images at 250 Hz of the horse's feet as it walked through an undistorted and calibrated capture volume (∼30 cm per cube edge) located on the forceplate.

Standard videos introduce geometric distortion to the images due to lens misalignments (*Dobbert*, *2005*) and the fluoroscopes create nonlinear pincushion and spatial artefacts (*Wang & Blackburn*, *2000*), which degrade image quality. To remove this distortion, we imaged a standardized sheet of perforated metal grid (1.6 mm thick with 3 mm diameter holes, spaced 4 mm apart in a 60° staggered pattern) (RS Components Ltd, UK) in front of the intensifiers and captured the grid images at 32 frames for 1 s. The exposure setting for

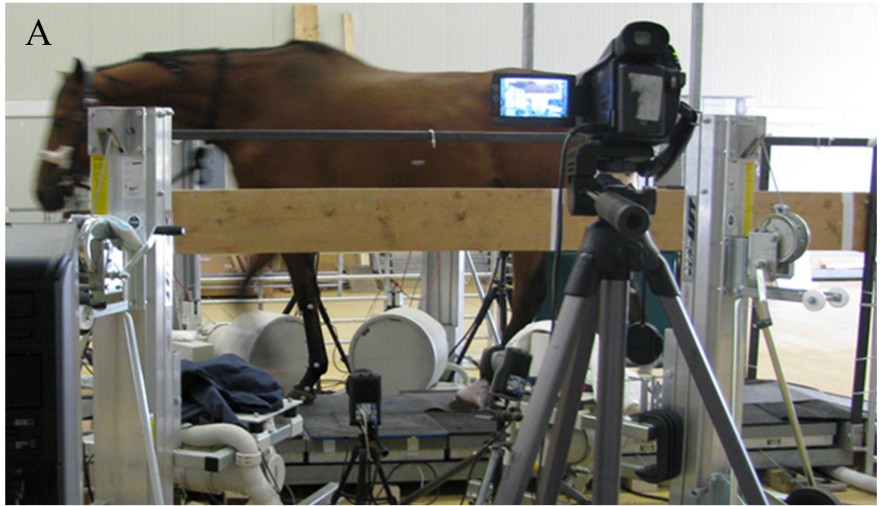

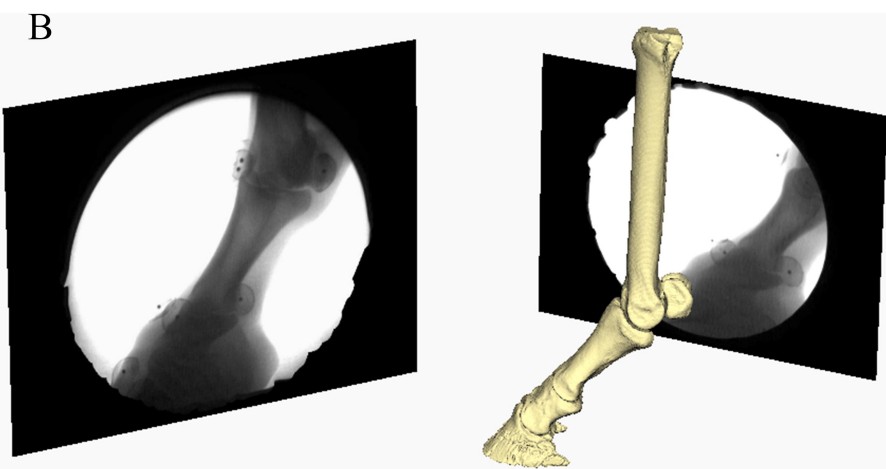

**Figure 1** **Image of experimental and virtual set-up.** (A) Experimental set-up of the horse walking on a custom-made platform retrofitted with a forceplate and surrounded by the bi-planar fluoroscopy system. (B) Virtual setup of the horse right forefoot based on the experimental alignment of the X-ray sources and the intensifiers. Images in black frames (right and left) illustrate the projections of the distal foot from the two X-ray cameras.

the two intensifiers during the collection of the grid images was respectively set at 57 kv, 38 mA and 50 kv, 19 mA.

To correct the distorted video images and the calibration cube images, we created undistortion files from the standardised reference grid images using the XrayProject MatLab (Mathworks Inc. Natick, Massachusetts, USA) protocol (www.xromm.org). By using the reference grid images of each intensifier as templates, we created a uniform set of squares and applied it to all video images (*Brainerd et al., 2010*). The distortion correction algorithm compared the spacing between holes in the fluoroscope image with the idealized spacing and calculated a transformation matrix for correcting all video images. We used a local weighted mean (LWM) distortion correction algorithm, implemented in MatLab.

To calibrate the video images and generate a DLT algorithm in order to create "virtual cameras" in the Maya virtual 3D space for rotoscopy of the horse's foot, we followed

the calibration process described by *Brainerd et al. (2010)*. We placed a custom-designed calibration cube with 64 steel 3 mm diameter spheres as calibration points within the field of view of the two intensifiers and recorded the calibration images at 32 Hz for 1 s (Fig. S1) (*Brainerd et al.*, *2010*). The calibration cube consisted of acrylic sheets of uniform thickness (5.42 mm), on which we drilled 16 holes of 3 mm diameter in a square pattern of 65 mm separation. Nylon pillars served as spacers between the tiers. The exposure settings for the two intensifiers during the collection of the calibration cube images were set at 46 kv, 39 mA and 71 kv, 11 mA.

The undistorted images of the calibration cubes for each camera were used as templates to calibrate all video images using protocols established by *Brainerd et al. (2010)*. The undistorted calibration cubes were digitised by manually selecting the centroid of a minimum of 12 calibration cube beads that could be seen in both cameras using MatLab. Digitizing was repeated until a calibration coefficient of low residuals (<0.3) was given for each camera. The resulting direct linear transformations (DLT) were used to optimise the focal point and axes of the cameras. The camera data were then imported to Maya software (Autodesk, San Rafael, California, USA) to recreate a virtual scene with similar coordinates to the experiment (*Brainerd et al.*, *2010*).

Exposure settings were set to 69 kV, 53 mA and 72 kV, 54 mA for the two sources, which were placed on the left side of the platform. Each intensifier was placed 2 m from its corresponding X-ray source on the right side of the platform such that the beams were oriented horizontally at approximately 90° to one another (Fig. 1A).

Kinetic data were collected simultaneously at a rate of 1,000 Hz using a forceplate (60 × 90 cm with Hall Effect sensors, 2,000 lb maximum vertical force; AMTI, Watertown, MA, USA). Prior to analysis, the forceplate data were low-pass filtered using a 4th order zero-lag Butterworth filter with a cutoff frequency of 15 Hz. All data were synchronized to the fluoroscope system.

The unshod horse was guided 344 times across the experimental platform over a period of two weeks. Following the end of the experiments for the unshod condition, the horse's forefeet received mild trimming and were each fitted with a stainless steel shoe with toe clips (5 inches wide) and 6 nails. The identical procedure was then followed to guide the shod horse over the platform 65 times on a subsequent day. The difference in trial numbers between the unshod (344 strides) and the shod (65 strides) conditions was due to the large number of spatially incomplete data for the former. Strides that were spatially incomplete (i.e., the right forefoot only stepped partially within the capture volume) and/or unsteady (i.e., with evident deceleration and acceleration following observation of the video images during data collection) were excluded from further analysis. Four steps from the shod and four steps from the unshod right forefoot that were spatially complete and steady were processed using the markerless XROMM (X-ray Reconstruction of Moving Morphology: *Brainerd et al.*, *2010*; *Gatesy et al.*, *2010*) workflow to construct a model and obtain 3D joint rotations and translations. The limited number of steps per conditions is a limitation of the XROMM approach when used in live animal studies for species as large as a Thoroughbred horse. For a step to be valid during the XROMM procedure, the animal has to step within the calibrated field of view without deviation. If there were minimal deviations from the

capture volume, we were unable to visualise the distal right forefoot in both cameras in order to extract the 3D joint kinematics.

## Model construction

The horse was euthanized at the end of the experiment for unrelated studies and its right forelimb was removed and frozen (–20 °C). Computed tomography scans (GE Lightspeed 16-detector unit; General Electric) were used to obtain the three dimensional (3D) skeletal geometry of the horse's forefoot (slice thickness 0.625 mm, 0.460 pixels mm$^{-1}$, 512 × 512 pixel images, 620 slices). These data were then processed to extract solid 3D polygonal mesh objects in Mimics (version 16.0; Materialise, Inc., Leuven, Belgium) and then imported into Maya (Autodesk, San Rafael, California, USA) to construct the biomechanical models' segments (Fig. 1B). Four segments were defined: the metacarpus (MC), first phalanx (P1), intermediate phalanx (P2) with the navicular bone and the distal phalanx (P3). An articulated skeleton was then created by hierarchically linking these segments (*Gatesy et al.*, *2010*) into a kinematic chain representing the metacarpophalangeal (MCP), proximal interphalangeal (PIP) and distal interphalangeal (DIP) joints (Fig. 2).

Joint orientations and positions were defined by first positioning all bone segments into a neutral anatomical pose (forefoot lying fully horizontally). Cylinders were then visually fit to the joint surfaces (i.e., the distal epiphyses of the MC, P1 and P2) to identify the axes of joint rotations. These locations were confirmed when manual manipulation of the virtual joint resulted in a natural motion where adjacent bones did not interpenetrate each other. Dissected cadaveric specimens and plastic models were used to further assess joint locations and positions. Transformations between coordinate systems were defined using a $Z$ (green axis), $Y'$ (yellow axis) and $X''$ (red axis) Cardan rotation, respectively representing abduction-adduction, flexion-extension and long axis rotation (Fig. 2). Axes were defined so that positive joint angles represented adduction, flexion and external long axis rotation.

## Markerless XROMM

Trajectories for each joint were quantified using established protocols (Brown University, USA; www.xromm.org) for scientific rotoscoping (markerless XROMM) (*Gatesy et al.*, *2010*; *Baier & Gatesy*, *2013*; *Baier, Gatesy & Dial*, *2013*; *Nyakature & Fischer*, *2010*). In brief, markerless XROMM is a technique that allows one to quantify 3D motion by animating model segments (i.e., 3D polygonal mesh objects) to match postures observed in experimental X-ray video images (Fig. 1B). For each experimental X-ray trial, the horse foot model was aligned with the bone X-ray silhouettes in undistorted and calibrated video images using the anatomical features of each bone as reference guides (Movies S1 and S2). Joint transformations (i.e., joint rotations and translations) were then extracted from the model for the MCP, PIP and DIP joints (Fig. 2). The MCP kinematic data were excluded from further analysis because the midshaft and proximal epiphysis were out of the field of view for most of the stride.

All steps for shod ($n = 4$) and unshod ($n = 4$) conditions were used to measure joint kinematics and foot kinetics, but a single representative step was selected for each condition

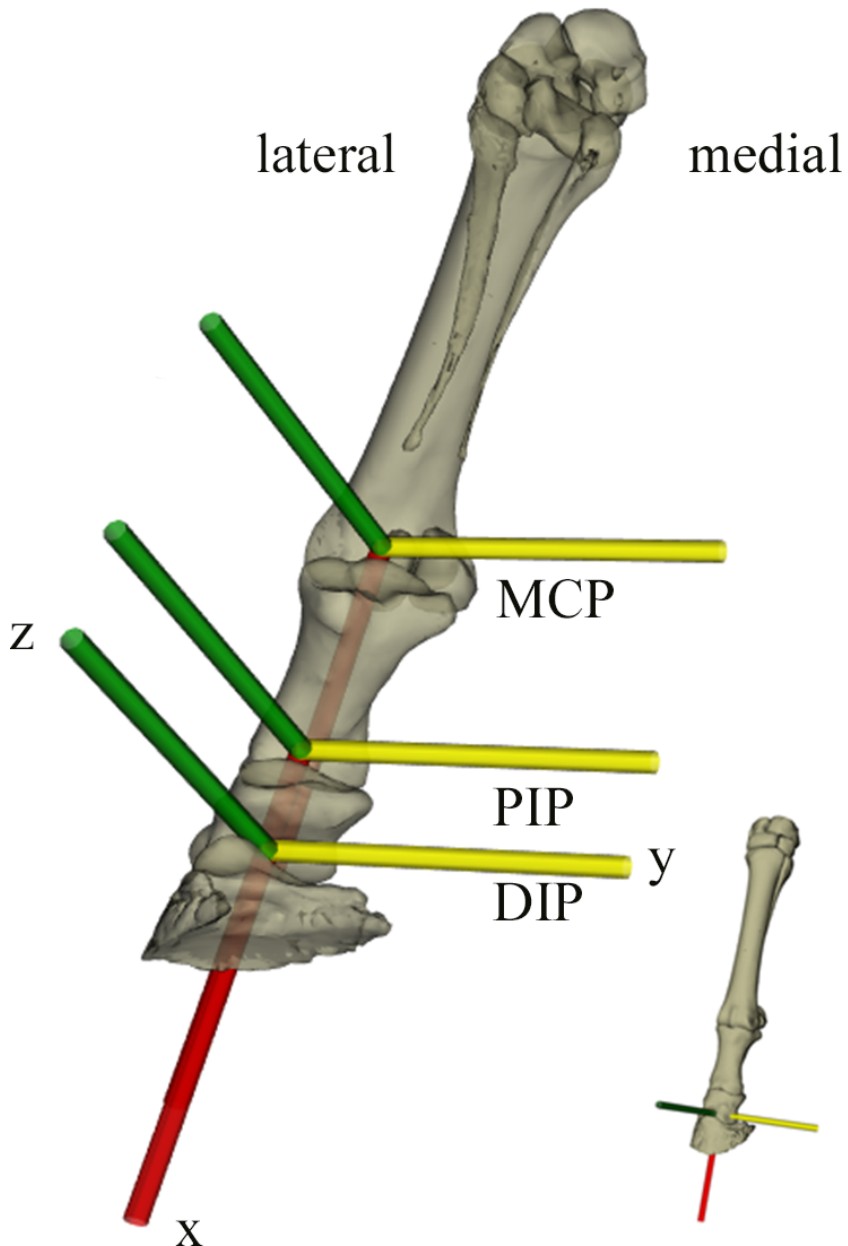

**Figure 2** **XROMM model with bone segments and coordinate systems for the metacarpophalangeal (MCP), proximal interphalangeal (PIP) and distal interphalangeal (DIP) joints.** Green, yellow and red arrows represent the $Z$, $Y'$ and $X''$ segment axes respectively.

(shod $n = 1$; unshod $n = 1$) for the subsequent mid-stance inverse dynamics calculations and FEA used to estimate bone stresses in the right forefoot digit.

### Inverse dynamics

Inverse dynamics methods were used to calculate the intersegmental forces (required for estimating bone stresses) at mid-stance for the two steps selected for FEA (shod and unshod conditions). Larger loads can occur at other points during the stance phase but the focus of this study is methodological and for simplicity we modelled solely mid-stance.

To perform the analysis, the skeletal model created for scientific rotoscoping was recreated in Software for Interactive Musculoskeletal Modeling (SIMM; Musculographics, California, USA). To create an exact replica of the original model, bone geometry was exported directly from Maya and imported into SIMM, where the geometry was reassembled by reproducing the original joint structure (i.e., number of joints and degrees of freedom). The model was then imported into OpenSim (*Delp et al.*, *2007*), which has a built-in routine to perform inverse dynamics.

Mid-stance was defined as the point halfway between foot strike and toe-off gait events, which were determined from the vertical GRF data. Mid-stance joint angles were exported directly from the XROMM workflow and used to position the model in OpenSim. Mid-stance GRFs, obtained from the synchronized forceplate data, were transformed into the same reference frame as the OpenSim foot model and then applied to the distal phalanx (P3). Data integrity between the motion and force data were verified by visually inspecting the location of the centre of pressure (CoP) (from forceplate data) relative to the foot placement (from XROMM kinematics) using OpenSim. Because the distal segments are small (i.e., low mass) and walking is a slow motion (i.e., low joint accelerations), gravity and inertial forces were assumed to be negligible relative to the forces resulting from the applied mid-stance GRFs. Thus, mass and inertial properties for all segments were not included in the analysis. OpenSim's inverse dynamics and joint reaction force routines were then used to calculate the intersegmental joint forces and moments acting on the segments at each joint. These data were expressed in the local frame of the segment and used as inputs into the FEA.

## Finite element analysis

For each phalanx (P1, P2 (including the navicular bone) and P3), a separate finite element model was created in Abaqus/CAE, software version 6.13 (Dassault Systemes Simulia Corp, Providence, Rhode Island, USA). The corresponding intersegmental forces from the inverse dynamics analysis were applied to each bone independently and stress was determined using the Abaqus/Standard implicit direct default solver.

### Bone models

Each 3D solid bone mesh representing a segment from the OpenSim model was imported into 3-Matic 9.0 software (Materialize Inc., Leuwen, Belgium) and converted into a volumetric mesh file of continuum linear tetrahedral elements of type C3D3. Each volumetric mesh file (preserving the coordinate systems of each segment as defined during the inverse dynamics analysis) was then imported into Abaqus/CAE 6.13 FEA software and converted into 10 node quadratic hybrid elements of type C3D10H. The element nominal size for all models was 2 mm. The P1, P2 (with the navicular) and P3 segments had 60,967; 42,401 and 35,725 elements respectively.

### Material properties

Due to a lack of specific material properties data for the bones of the distal foot of horses, linear elasticity, homogeneity and isotropy were assumed for each bone model. Assumptions regarding isotropy and homogeneity should create a constant error between our models

and thus do not influence bone stress comparisons between the shod and the unshod horse. We assigned a Young's modulus (E) value of 16,000 MPa and Poisson's ratio (v) of 0.3 to the P1 and P2. The P3 in horses consists of dense trabeculae and was thus assigned a modulus of 10,000 MPa and Poisson's ratio of 0.3 (*Rho et al.*, *2001*; *Jansová et al.*, *2015*).

### Loads and constraints

To load each bone model, we applied the intersegmental forces calculated during the inverse dynamics routine at mid-stance to the bone segment surface associated with the joint of interest. To apply each load, we selected the nodes at the entire joint surface for P1, P2 and P3 and divided the $x, y, z$ force components uniformly across the joint surface.

Each bone model was loaded and analysed separately, using the intersegmental forces to approximate the loads arising from skeletal geometry, kinematics and kinetics. The P1 bone model was loaded at the MCP joint (Fig. S2). The P2 bone model (with the attached navicular) was loaded at the PIP joint (Fig. S2) and the P3 bone was loaded at the DIP joint (Fig. S2). The proximal and intermediate phalanges were constrained by selecting the entire joint surface at the distal end of the bones and fixing rotations and displacements about all axes. The P3 was constrained on the most cranial point of the palmar part.

To test the effect of constraints on bone stresses, we conducted a sensitivity analysis on the P1 and P3 of the shod condition. The P1 was fixed to be constrained on three nodes at a location approximated as the centroid of the resultant loads (to withstand bending). The rest of the nodes on the distal joint surface were fixed solely on the direction along the long axis of the bone (to withstand compression). The P3 constraints were varied by increasing the surface area of the constrained nodes and fixing the rotations and displacements about all axes.

We visually compared von Mises, maximum principal stress (S1) and minimum principal stress (S3) patterns between the shod and unshod conditions for the P1, P2 and the navicular bone, and P3.

### Data analysis

Stance phase kinetic and kinematic data were normalised to 100% stance phase duration (i.e., ground contact time). Descriptive statistics were used to quantify the walking speed within the shod ($n = 4$) and the unshod ($n = 4$) conditions. A cross-correlation analysis was conducted to assess the correlation between the unshod and shod horse in overall mean GRF patterns (in the craniocaudal, mediolateral and vertical directions) and the overall pattern in mean angle of flexion-extension for the PIP and DIP joints across the stance phase (10–90%). The first and last 10% of the stance phase were excluded from the kinematics data due to noise caused during markerless XROMM analysis. Due to the small sample size (1 horse and four trials per condition), the differences in the kinematics, kinetics and bone stresses were not assessed for statistical significance.

## RESULTS

### Speed data

The mean walking speed of the shod and unshod conditions was at 0.72 ms$^{-1}$ and 0.76 ms$^{-1}$ respectively (Table 1). This corresponded to a Froude number—which is a dimensionless

**Table 1** Minimum (Min), maximum (Max) and mean Froude number and velocity data, with standard deviation (SD), for the shod ($n = 4$) and unshod ($n = 4$) horse trials.

| Condition | $n$ | Min | Max | Mean | SD |
|---|---|---|---|---|---|
| **Froude number** | | | | | |
| SHOD | 4 | 0.045 | 0.076 | 0.058 | 0.013 |
| UNSHOD | 4 | 0.060 | 0.069 | 0.064 | 0.0045 |
| **Velocity (ms$^{-1}$)** | | | | | |
| SHOD | 4 | 0.64 | 0.83 | 0.72 | 0.079 |
| UNSHOD | 4 | 0.74 | 0.79 | 0.76 | 0.024 |

representation of movement speed (*Alexander & Jayes, 1983*; $Fr = velocity^2 * (9.81\ ms^{-2} * hip\ height)^{-1}$)—of 0.05 for the shod condition and 0.06 for the unshod condition (Table 1), indicative of a slow walk. The footfall patterns also maintained the typical lateral sequence found in walking (Data S1).

## Kinetic data

The GRF data from the shod ($n = 4$) and unshod ($n = 4$) conditions during the stance phase of locomotion are shown in Fig. 3 and Data S2. In all directions the force pattern was quite similar between the shod and unshod conditions (Fig. 3). The results from the cross-correlation analysis assessing the correlation between the unshod and shod horse in overall GRF patterns showed that there was a high positive correlation between the craniocaudal GRF patterns (with a maximum correlation coefficient of 0.994 with a 2% lag of the shod pattern). These results also showed a high positive correlation between the vertical GRF patterns (with a maximum correlation coefficient of 0.996 with a 0% lag of the shod pattern). While there also seemed to be a large similarity in mediolateral GRF patterns, the strength of the positive correlation was lower than in the other directions (with a maximum correlation coefficient of 0.747 with an 11% lag of the shod pattern).

The craniocaudal GRF at the beginning of the stance phase was directed caudally, which then shifted cranially from mid-stance until the end of the stance phase of locomotion (Fig. 3). The maximum cranial GRF for the shod horse was shown at approximately 75% of stance (497 N) and for the unshod horse at 78% of stance (396 N). Between 75–78% of stance, the shod horse showed on average a 21% higher craniocaudal GRF than the unshod horse, yet this difference was not assessed for significance.

The mediolateral GRF for the shod condition was directed medially throughout the whole stance phase and reached an average maximum force of 85 N after mid-stance. Contrastingly, the mediolateral GRF for the unshod horse was directed medially during the first 10% of the stance phase, shifted laterally until late mid-stance, and then was redirected medially until the end of the stance phase. The highest mediolateral GRF for the unshod horse was laterally directed and occurred before mid-stance (~40–45%), reaching approximately 100 N.

There was a strong similarity in the vertical GRF pattern between the shod and unshod conditions during the stance phase. However, at mid-stance, the vertical GRF of the shod condition (3,195 N) was approximately 10% higher than the unshod condition (2,888 N).

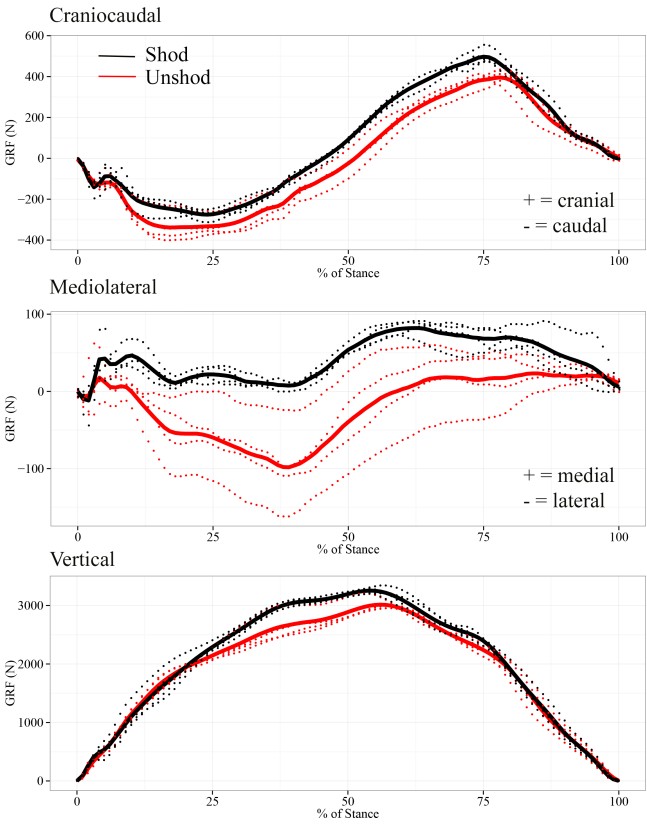

**Figure 3 Ground reaction forces normalised to 100% stance phase for the shod (black lines) and unshod horse (red lines).** For the craniocaudal GRF, cranial and caudal are positive and negative respectively. For the mediolateral GRF, medial is positive and lateral is negative. Solid lines represent the trials used in the subsequent finite element analysis.

## Kinematic data

The kinematic data for the shod ($n = 4$) and unshod ($n = 4$) conditions during the stance phase of locomotion are shown in Fig. 4 and Data S3. The results from the cross-correlation analysis between the unshod and shod conditions showed a high positive correlation for the PIP mean joint angle (with a maximum correlation coefficient of 0.989 with a 0% lag of the shod pattern). A strong positive correlation was also found for the DIP mean joint angle (with a maximum correlation coefficient of 0.975 with a 0% lag of the shod pattern).

We describe some differences in kinematic patterns here for the shod vs. unshod conditions but it is very important to note that none of these have been tested for true statistical significance, because of the small sample sizes. Overall, in both the shod and unshod conditions, the PIP joint was in a flexed posture early in the stance phase (1040%) and ended in an extended posture before mid-stance (45% of stance) until the end of the stance phase (90%). Between 10–90% of the stance phase, the PIP joint for the shod condition had an average range of motion (ROM) of 14 degrees. The average ROM for the unshod condition's PIP joint was estimated at an average of 13 degrees.

The maximum difference between the DIP joint angle in the shod and unshod conditions was at 40% of stance, when the unshod horse DIP approximates a neutral posture ($\sim = 0°$),

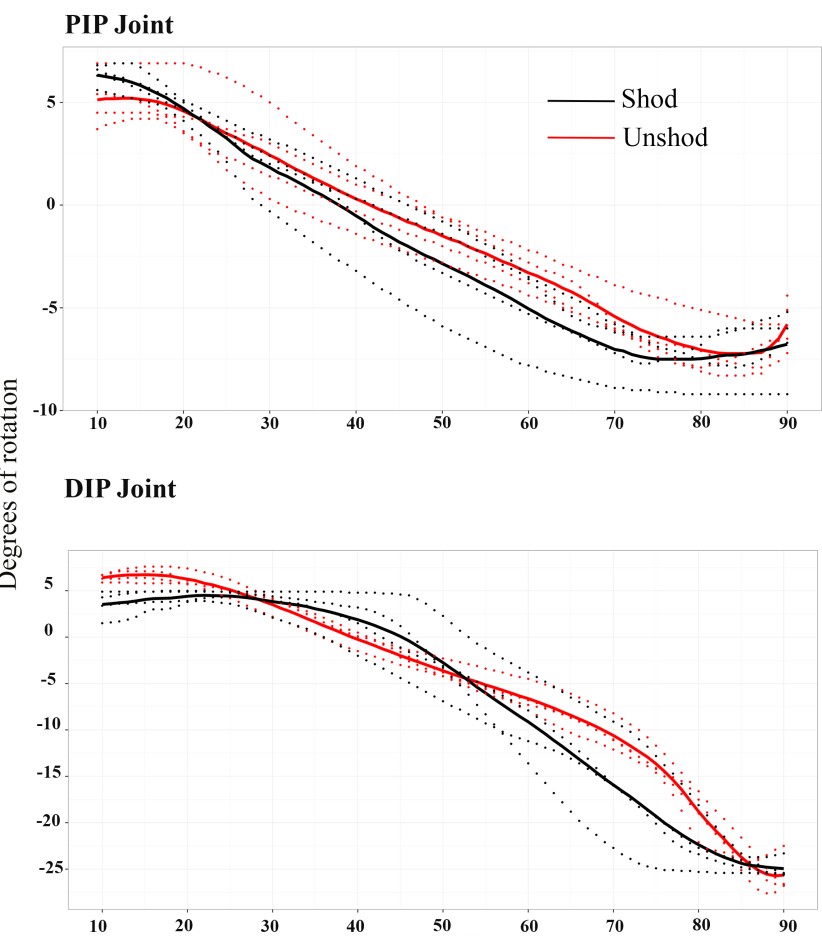

**Figure 4  Kinematic data.** Degrees of rotation for the proximal interphalangeal (PIP) and distal interpha-langeal (DIP) joints, around the flexion (positive)—extension (negative) axes during the stance phase for the shod (black line) and the unshod (red line) conditions. Dotted lines show the individual trials and the bold lines show the mean degrees of rotation for each condition.

while the shod horse DIP exhibited a more flexed posture. At mid-stance and until the end of the stance phase, the DIP extended in both the shod and unshod conditions (Table 2).

## Finite element analysis (FEA)

The intersegmental forces obtained from the inverse dynamics technique and applied to the different segment models (P1, P2 and P3) during the FEA are presented in Table 3. Our FE analysis showed that the shoe increased the concentration of von Mises and maximum principal stresses on the dorsal (Fig. 5 and Fig. S3) and palmar (Fig. 6 and Fig. S4) aspects of the distal (P1, P2, P3) bones of the horse's forefoot; however this possible increase has not been assessed for significance due to the small sample size. Comparisons of the principal stress patterns showed that in both conditions the dorsal (Figs. S3 and S5) and palmar aspects (Figs. S4 and S6) of all phalanges respectively underwent tension and compression.

**Table 2  Mean degrees of rotation for the proximal interphalangeal (PIP) and distal interphalangeal (DIP) joints of the shod and unshod conditions during the stance phase.** Note that none of these differences can be considered statistically significant.

| % Stance | PIP | | DIP | |
|---|---|---|---|---|
| | UNSHOD | SHOD | UNSHOD | SHOD |
| 10 | 5.1 | 6.3 | 6.4 | 3.5 |
| 15 | 5.2 | 5.8 | 6.7 | 4.1 |
| 20 | 4.6 | 4.7 | 6.3 | 4.4 |
| 25 | 3.5 | 3.3 | 5.1 | 4.4 |
| 30 | 2.4 | 1.8 | 3.5 | 3.8 |
| 35 | 1.3 | 0.7 | 1.6 | 3.1 |
| 40 | 0.3 | −0.5 | −0.3 | 1.9 |
| 45 | −0.6 | −1.8 | −2 | 0.1 |
| 50 | −1.5 | −2.9 | −3.6 | −2.8 |
| 55 | −2.4 | −3.9 | −5.2 | −6 |
| 60 | −3.3 | −5.1 | −6.6 | −9.1 |
| 65 | −4.2 | −6.1 | −8.4 | −12.5 |
| 70 | −5.4 | −7 | −10.6 | −15.9 |
| 75 | −6.4 | −7.5 | −13.7 | −19.3 |
| 80 | −7.1 | −7.5 | −18.8 | −22.4 |
| 85 | −7.2 | −7.3 | −23.7 | −24.4 |
| 90 | −5.8 | −6.8 | −25.7 | −25 |

**Table 3  Intersegmental forces in Newtons (N) assigned to the shod and unshod horse finite element models.** Positive values represent forces applied in the distal (i.e., compressive), lateral and cranial directions.

| Force | Proximal-distal | | Medial-lateral | | Cranial-caudal | |
|---|---|---|---|---|---|---|
| | Shod | Unshod | Shod | Unshod | Shod | Unshod |
| P1 | 2,750 | 2,413 | −1,673 | −1,394 | 45 | 198 |
| P2 | 2,503 | 2,354 | −2,024 | −1,492 | 48 | 199 |
| P3 | 2,580 | 2,293 | −1,924 | −1,583 | 47 | 200 |

The sensitivity analysis on the constraints for the P1 showed that fixing the constraints on the distal joint surface reduced von Mises stresses compared to fixing the centroid nodes and constraining the rest of the nodes to align with the long axis of the bone. However, in both cases bending occurred craniocaudally (Fig. S7).

Our sensitivity study of the P3 constraints showed that an increase in the surface area of the constrained nodes reduced von Mises stresses dorsally (Fig. S8).

## DISCUSSION

Our study implemented a new methodology that combines XROMM, inverse dynamics methods and FEA to quantify the effect of wearing a stainless steel shoe on the biomechanics of the right forefoot of a Thoroughbred horse during slow walking *in vivo*, although

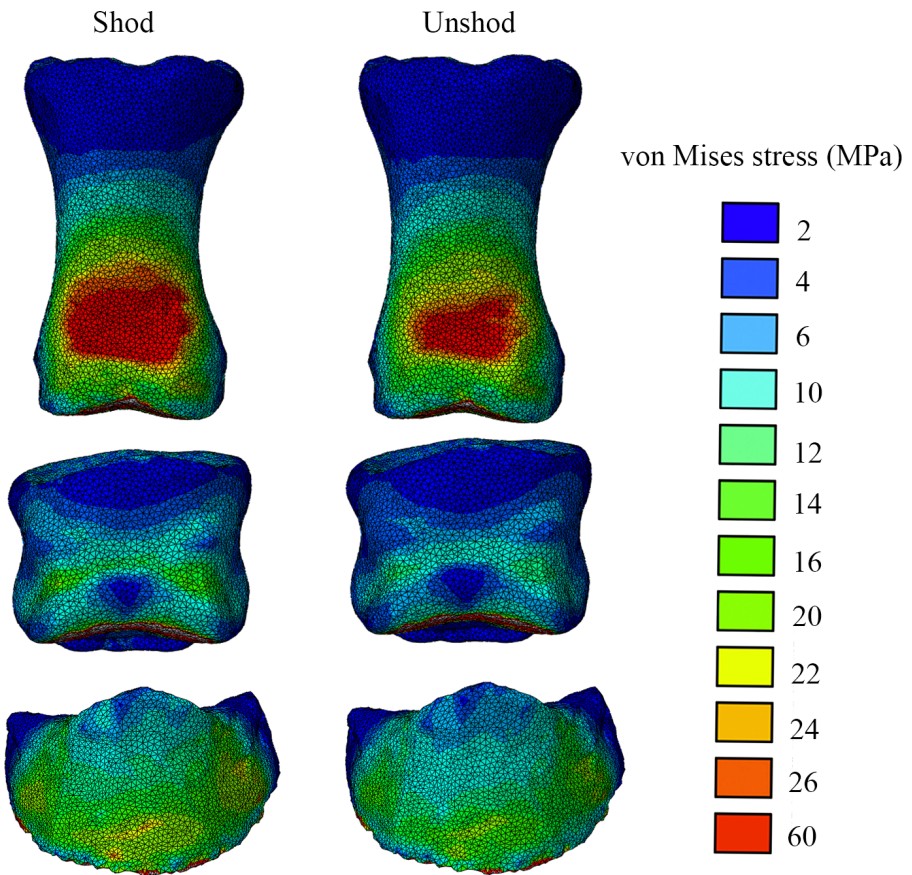

Shod      Unshod

von Mises stress (MPa)

| | |
|---|---|
| | 2 |
| | 4 |
| | 6 |
| | 10 |
| | 12 |
| | 14 |
| | 16 |
| | 20 |
| | 22 |
| | 24 |
| | 26 |
| | 60 |

**Figure 5** **Von Mises stress (MPa) distribution results for the shod and the unshod horse foot, in dorsal view.** Bones shown from top to bottom are the P1, P2 and P3. Warm (red) and cold (blue) colours show higher and lower von Mises stresses respectively.

admittedly our small sample sizes preclude conclusive detection of any statistically significant differences.

Our kinetic analysis showed that the stainless steel shoe may shift craniocaudal, mediolateral and vertical GRFs over much of stance, which is in accord with previous studies in Thoroughbred (*Roepstorff, Johnston & Drevemo*, *1999*) and Warmblood horses (*Willemen, Savelberg & Barneveld*, *1998*). The reported differences in GRFs between the shod and unshod horse may be due to the grip or impact attenuation properties of the shoe material. Previous studies have reported that horseshoe materials have variable frictional and damping properties and can affect the dynamics of the foot in horses (*Heidt et al.*, *1996*; *Wilson et al.*, *1992*; *Pardoe et al.*, *2001*). It is thus possible that an increase in the craniocaudal GRF may be due to the gripping properties of the steel shoe when in contact with the experimental platform, which could shorten the slip time and increase musculoskeletal forces after impact (*Willemen*, *1997*; *Johnston et al.*, *1995*).

Using XROMM we accurately measured *in vivo* joint kinematics data of the Thoroughbred horse under shod and unshod conditions without artefacts from the skin and other underlying soft tissue motions. The results were consistent with our expectations

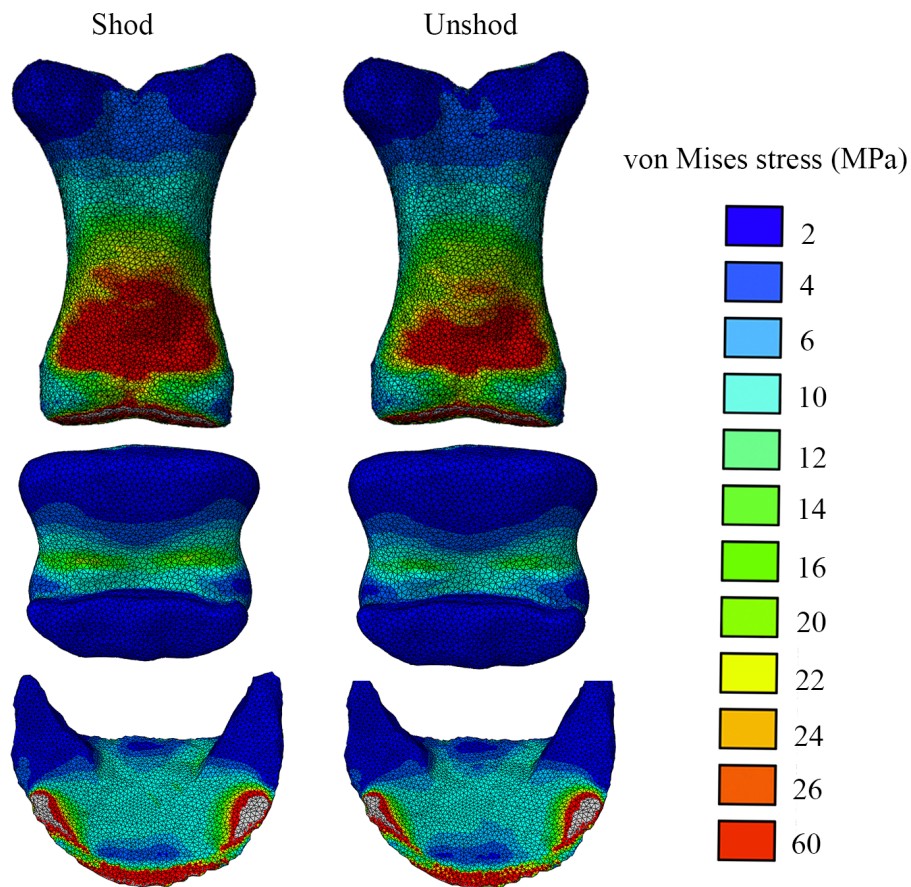

Shod          Unshod

von Mises stress (MPa)

| | |
|---|---|
| ■ | 2 |
| ■ | 4 |
| ■ | 6 |
| ■ | 10 |
| ■ | 12 |
| ■ | 14 |
| ■ | 16 |
| ■ | 20 |
| ■ | 22 |
| ■ | 24 |
| ■ | 26 |
| ■ | 60 |

**Figure 6  Von Mises stress (MPa) distribution results for the shod and the unshod horse foot, in palmar view.** Bones shown from top to bottom are the P1, P2 and P3. Warm (red) and cold (blue) colours show higher and lower von Mises stresses respectively.

for a cursorial animal such as our horse subject. During the stance phase in both shod and unshod conditions, the horse's forefoot joints extended by large amounts but minimal motion occurred in adduction-abduction and longitudinal rotation. This finding corresponds to those from previous kinematic studies on unshod horses during walking that also reported flexion-extension as the dominant rotation and only minimal adduction-abduction and longitudinal rotations (*Clayton et al.*, *2007a*; *Clayton et al.*, *2007b*). The negligible rotational differences between the shod and the unshod conditions during longitudinal rotation and adduction-abduction found in our study likely are confounded not only by our small sample sizes but also by noise due to the very small rotations and human error in rotoscoping such fine details of motion. *Menegaz et al.*'s (*2015*) kinematic study on pig feeding also attributed minimal rotations that failed to pass their precision threshold to noise introduced by the XROMM analysis procedure. An additional issue of uncertainty with our subject is whether it had any foot balance issues, either in terms of clinical observations or centre of pressure patterns, but available information do not allow us to assess this possibility.

Our kinematic data for both the shod and unshod conditions showed flexed postures in both the PIP and DIP joints from approximately 10% of stance until mid-stance with maximal joint extension occurring just before the foot leaves the ground. This finding is consistent with previous studies of horse foot kinematics in both walking and trotting, which have shown that the PIP and DIP joints maintain a similar motion pattern in those gaits, with changes evident only in the amounts of rotation (*Chateau, Degueurce & Denoix*, *2004*; *Clayton et al.*, *2007b*). The greater extension of the DIP joint relative to the PIP joint after mid-stance and at late stance is in accord with previous research on walking and trotting horses in shod and unshod conditions (*Clayton et al.*, *2007b*; *Roach et al.*, *2015*; *Roepstorff, Johnston & Drevemo*, *1999*). Although the horse received training prior to data collection, walking on a platform surrounded by equipment could have intimidated the animal and thereby influenced its natural locomotor behaviour and speed. Future research should measure more individuals to account for intraspecific variations in locomotor behaviours.

A constraint on this sort of multi-individual study, however, is that each individual must have its distal limb CT or MRI scanned to obtain subject-specific morphological data for XROMM analysis, which requires mild sedation, anaesthesia or euthanasia, with accompanying ethical dilemmas and risks, in addition to the very time-intensive nature of not only collecting synchronised kinematic and kinetic data but also processing the XROMM data and subject-specific musculoskeletal (e.g., OpenSim) and FEA modelling analyses. For this study, we also simplified our implementation of the inverse dynamics technique by assuming a static mid-stance (i.e., no segment accelerations) posture and that individual foot segments were massless. A sensitivity analysis showed that these assumptions had a negligible (<1%) effect on the calculated intersegmental forces and joint torques used in the FEA. However studies of different species and/or motions may invalidate these assumptions and require collecting additional detailed segment inertia and mass properties. Regardless, the inverse dynamics approach presented here can easily be generalized to account for segment accelerations and gravity, if necessary. Hence, despite our study's restriction to measuring a few steps of one individual and assuming a static posture and massless foot segments, it is an important example of the integration of 3D biomechanical methods and their application to fundamental problems in equine locomotion, care and welfare.

Our FEA results revealed an increase in the von Mises and principal bone stress magnitudes in the shod (vs. unshod) horse's forefoot phalanges at mid-stance. Both conditions showed increased stresses on the distal epiphysis of the proximal phalanx in the dorsal and ventral view (Figs. 5 and 6). The unshod horse showed slightly higher stresses than the shod horse around the sagittal groove of the P1, yet stresses around this area were low compared to the midshaft and the proximal epiphysis. This finding is partly similar to the findings of *O'Hare et al.* (*2013*) for walking, yet their study found higher stresses around the sagittal groove of the proximal phalanx. This is potentially due the fact that the proximal-distal force that was assigned to the *O'Hare et al.* (*2013*) model to simulate walking was 3,600 N, but our inverse dynamics analysis results were 2,503 N and 2,354 N for the shod and unshod conditions respectively. In addition our model is solid, stiffer and thus will have smaller responses to stress.

Our FE analysis also showed an increase in von Mises and principal stresses around the midshaft of the intermediate phalanx and around the proximal borders of the navicular bone. However, in both the shod and unshod conditions, stresses around the navicular bone were minimal. A more advanced model that includes the superficial and deep digital flexor tendons is required to assess whether the navicular bone acts more like a lever for the deep digital flexor tendon (DDFT) (*Eliashar, McGuigan & Wilson, 2004*), rather than in direct load-bearing.

A major issue in subject-specific FE models when based on *in vivo* experiments is the location of the constraints on the bones. We here showed that expansion of the surface area of the constrained nodes on the palmar aspect of the P3 reduced the magnitudes and patterns of stresses dorsally but it did not affect the overall comparisons between the shod and unshod conditions. In all cases, the shod horse showed higher concentration of stresses on the P3 than the unshod condition, with the caveat that none of these differences were statistically tested.

Boundary conditions are an area of uncertainty in FEA studies of long and short bones when *in vivo* conditions are simulated. In this study we applied our loads to the entire surface of the joint of interest and we constrained all nodes on the distal joint surface for the P1 and P2. Our approach may have over-constrained the models, but is reproducible when FEA is conducted on bones of varying shape and size. An alternative approach would be to only constrain the node that is located along the same centroid axis with the load centroid, and constrain the rest of the nodes only on the long axis of the bone. This modelling approach would also allow compression whilst preventing bending but it is not known if, biologically, the joints are loaded and constrained along their long axes so precisely. It is also not feasible to define a node on the exact centroid axes on a mesh file of a bone when the geometry is not symmetrical. Our sensitivity analysis of the boundary conditions applied to the P1 showed that our modelling approach reduces stresses on the distal end of the P1, but in both cases bending occurred craniocaudally (Fig. S7).

FEA is a modelling approach and inevitably will introduce modelling errors/artefacts. In a comparative context (e.g., shod vs. unshod) it is unlikely that modelling artefacts will affect the ultimate comparisons, however rigorous sensitivity analyses are recommended for clinical applications when absolute stress magnitudes are essential.

There is also the valid concern that, although our experimental data (kinematics and kinetics) are *in vivo* measurements of real motions and have a high degree of precision, our OpenSim (i.e., inverse dynamics) and FEA modelling analyses did not account for the tissues of the hooves themselves, the shoes, ligaments, tendons, frog or other soft tissues that would certainly alter the mechanics of the foot. Thus our analysis shows what the influence of shod vs. unshod conditions of our horse subject were solely upon the *in vivo* dynamics (including the altered GRFs and motions) and upon the stresses within the bones in the theoretical case of those bones bearing all loads themselves. Certainly the absolute values of the stresses would change with the addition of soft tissue data and neuromuscular control, but it is less certain how much the relative stresses would change between the shod vs. unshod conditions. Regardless, this will remain unknown until more sophisticated models are created and additional studies are conducted. Even so, we have presented the

first analysis that integrates state-of-the-art methods for kinematic and kinetic analysis with musculoskeletal modelling and finite element analysis methods for the distal foot of horses, which itself is a considerable methodological advance that future studies can build upon.

Our preliminary study illustrates that the stainless steel shoe may influence the dynamics and mechanics of a Thoroughbred horse's forefoot during slow walking, although our results are inconclusive in some important aspects. Certainly, more research is needed to quantify the effect of the shoe on the equine forefoot during the whole stance phase, under different trimming protocols, at faster speeds/gaits and with more individuals and strides as well as a similar focus on the hindfeet. Expansion of this research question, especially via the application of this novel combination of *in vivo* experiments and computer models should not only create a foundation of stronger data and inferences on which future studies can continue to build on, but can also bolster confidence in equine biomechanics to better understand the form, function and pathological relationships of the anatomical tissues of the equine foot.

## ACKNOWLEDGEMENTS

We thank Sharon Warner, Emil Olsen, Renate Weller, Luis Lamas, Emily Sparkes, Heather Paxton and Julia Molnar for their assistance and technical support during data collection. We also thank Phil Pickering for his assistance with the setup of the custom-made platform and the fluoroscopy system and Justin Perkins for allowing us access to the Thoroughbred horse. We thank Jan Janzekovic and Hyab Mehari Abraha from the Moving Morphology & Functional Mechanics Laboratory for their assistance in data analysis. We are grateful to Sandra Shefelbine and Andrea Pereira for useful discussions on FEA and our colleagues at the University of Brown (Sabine Moritz, David Baier and Beth Brainerd) for their endless support during the processing of the XROMM data. Particular thanks are also due to Todd Pataky, Renate Weller and Vivian Allen for valuable discussions. Finally, we appreciate the constructive criticisms of our Editor and our 3 reviewers.

### Funding

Financial support received from a Biotechnology and Biological Sciences Research Council (BBSRC) Project Grant (BB/H002782/1) to JRH and Renate Weller. The funders had no role in study design, data collection and analysis, decision to publish, or preparation of the manuscript.

### Grant Disclosures

The following grant information was disclosed by the authors:
Biotechnology and Biological Sciences Research Council (BBSRC): Project Grant (BB/H002782/1).

### Competing Interests

John R. Hutchinson is an Academic Editor for PeerJ.

## Author Contributions

- Olga Panagiotopoulou conceived and designed the experiments, performed the experiments, analyzed the data, wrote the paper, prepared figures and/or tables, reviewed drafts of the paper.
- Jeffery W. Rankin conceived and designed the experiments, performed the experiments, analyzed the data, prepared figures and/or tables, reviewed drafts of the paper.
- Stephen M. Gatesy analyzed the data, contributed reagents/materials/analysis tools, prepared figures and/or tables, reviewed drafts of the paper.
- John R. Hutchinson conceived and designed the experiments, reviewed drafts of the paper.

## Animal Ethics

The following information was supplied relating to ethical approvals (i.e., approving body and any reference numbers):

The study was reviewed and approved by the Royal Veterinary College's Ethics and Welfare Committee (approval number URN 2011 1094).

## Data Availability

All raw data are uploaded as Data S1–S3.

## Supplemental Information

Supplemental information for this article can be found online at http://dx.doi.org/10.7717/peerj.2164#supplemental-information.

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
