# Peer review of "A preliminary case study of the effect of shoe-wearing on the biomechanics of a horse’s foot"

_PeerJ, doi:10.7717/peerj.2164_

## Round 0.1 · original submission · Major Revisions

I think this is a great study, and one that is definitely worth publishing. I've given it 'major revisions' because that is what 2 out of 3 reviewers suggested, but you may find it doesn't take too long to address their comments. The major revision, flagged by both Reviewer 2 and 3, is to refocus the manuscript first and foremost as a proof of concept of the novel method, with less emphasis on the interpretation of the results. I think this is a good suggestion. Reviewer 3 also has a number of specific queries about the methodology, specifically the application of boundary conditions.

·

Basic reporting

Movie S2 not working.
Figure 7: Which von Mises stress values are in Figure - if mean, it could be written in description.
Supplementary Figure S2: The constrains and the loads could be highlighted in different colour for easier identification.

Experimental design

Table 3: How were these values chosen? In which direction they were applied?
line 343 - Chapter FEA: The bones were modelled and loaded separately?

Validity of the findings

The soft tissues are necessary in order to obtain realistic data. The authors could mention in abstract that the FE model includes only bones so that the reader knows what to expect from reading the paper.

Additional comments

Really interesting new method of combining 3D data from XROMM, inverse dynamics, and FE analysis on horse's foot. It surely required huge amount of work.

Reviewer 2 ·

Basic reporting

The basic reporting and the written style of the paper is very good. There are a few sentences that are not particularly clear but I have pointed these out.
There are a few references that have not been cited that might be of interest to the authors and relevant to this manuscript. There are a series of articles by Harrison, Whitton et al. (2012 and 2014) that describe the production of a subject specific computational model of the equine distal limb. These manuscripts are relevant for the discussion of the current work.
The figures and tables included are clear, but since the tables include data that was not statistically different between the shod and unshod groups, the authors might be able to decrease the number of tables included.

Experimental design

The authors describe a novel use of XROMM in the equine distal limb. This has potential to be a useful research tool but currently has limitations which are addressed by the authors. The novelty of the application to the horse is the important part of this manuscript and is somewhat overshadowed but the efforts to discuss the descriptive findings of the study (see validity of findings section). In essence, this is a pilot study and a proof of concept manuscript and it should be badged as such.
The authors are very thorough in their presentation of the experimental design. I have a few questions regarding the production of the FE model (as a non-expert in this area) which will require someone more expert about the technique to address. The queries are relevant to the statements in lines 196-200, 257 – 271, and the assigned forces in Table 3 – which relate to construction of the model and the assignment of the material properties.

Validity of the findings

As regards the actual experimental data, the comparison between the shod and the unshod foot and the effect of shoeing on GRF, joint angles and von Mises stresses there was no statistical difference between the two conditions. The authors spend a lot of time talking about the differences between the two conditions when this is simply descriptive and related to one horse and 4 trials of that horse (joint angles and GRF) or one horse (FE model). Therefore, I found the findings of the experimental data overstated and their significance to the study of the biomechanics of the foot when comparing shod v unshod horses not relevant (since no statistical difference was seen). I think that the experimental design and the XROMM technique has potential to be extremely useful for studying the equine digit, but that should be the focus of the manuscript, not the differences between the conditions.

Annotated reviews are not available for download in order to protect the identity of reviewers who chose to remain anonymous.

Reviewer 3 ·

Basic reporting

No comments

Experimental design

More details are needed for FEA model.

Validity of the findings

The boundary conditions of the FEA are likely significantly affecting the results. It is unclear why the kinematics and kinetics are not significantly different, yet the results of the FEA are.

Additional comments

A preliminary case study of the effect of shoe-wearing on the biomechanics of a horse’s foot
This study combines XROMM motion data and kinetics in a musculoskeletal model of the horse foot to determine joint loads. These loads are used in finite element models to determine the difference between shod and unshod stresses.
General:
The authors are congratulated on a significant amount of work that combines many state-of-the-art techniques. Overall the workflow is excellent. Some of the methods are not clear (as detailed below). The authors conclude that shoeing increases stresses. They are encouraged to describe more clearly what exactly is contributing to the difference in stresses. They show kinetics and kinematics which are very similar. The only large difference is the mediolateral GRF. However, they say that the motion and GRF results are not significantly different (likely because of the small sample size. If the geometry, material properties, motion, and forces are not different, it is unclear how the stresses can be significantly different. The authors should explore more thoroughly what is causing the difference in stresses and if the stresses really are significant given the uncertainty of the loads and boundary conditions.
There are significant concerns about how the boundary conditions are applied on the segments. See below.
Specific
1. Methods – For the most part, well described. Check supplemental movie S2. It does not show anything (at least the one that was uploaded).
2. Methods – Inverse dynamics: Can you clarify that you are just doing inverse dynamics. With no muscles, Open Sim is not even really necessary. You are just solving the kinematic equations? Are you using moment of inertia (I) from your scanned bones? Are the bones hollow or solid filled when you import them? Were joint loads only solved at mid-stance? Why?
3. Methods – FEA: Bone geometry – Were the meshes hollow or solid?
4. Method – FEA Loads: Specify which intersegmental forces were applied. Only midstance? Why? Because this is where the resultant ground reaction force is maximum? Did you confirm that joint forces are maximum here as well?
5. Methods FEA Loads: Clarify how the loads were applied. In Figure S2 it appears that there is a point force distributed across the entire surface of the joint? The sum of which equals the calculated joint reaction force?
6. Methods FEA Loads: How you have applied your boundary conditions will impose bending because you are loading it like a cantilever. Having a completely fixed end, particularly in such short bones will induce artifact. Did you check that the resultant centroid aligns with the load centroid? This would ensure that you are not creating bending just because of your boundary conditions. It would be better to constrain only the node at the distal end that is closest to the location of the z/y location of the loading centroid. The rest of the nodes could be constrained only in the x direction.
7. Is P3 the hoof? Did you model the shoe? If not, why not? The shoe has much different propreties than the bone and is likely to make a big difference in stresses in P3.
8. Why is only the outer border of P3 constrained?
9. The analysis lumps all elements in the middle together to calculate the average VM stress. Why? This defeats the purpose of FEA! It is better to compare the patterns of stress, maximum stresses, regions of maximum stresses. You typically do not do a subject specific FEA to get statistical results.
10. Results: Kinetic data. Either Figure 3 is mislabeled or the description of the results is backwards.
11. line 316: Specify that the highest load is in the lateral direction as this is opposite from the other condition.
12. Figure 3 – specify on the y axis which direction is which. (e.g. + = medial, - = lateral)
13. Kinematic data (332) The shod conditions was 45% higher. Perhaps this is true, but only because the numbers are so small! Better to report the total range of motion for each rather than as a percentage.
14. line 338: DIP joint angle extended in shod, flexed in unshod. Again this might be true but both are very close to zero. Do you think it is a significant point?
15. Analysis of the FEA by lumping elements, calculating mean and seeing if it is significantly different to between shod/unshod is inappropriate. It is better to compare the patterns of stresses (but also unclear how these will differ when you say that kinematics and kinetics were not significantly different). Explore more why you are getting different stresses. Be aware that you boundary conditions play a significant role in the stresses you are getting.
16. line 357 – Do the GRF really increase? You say in results that is it not significant.
17. Reduce the amount of ‘whilst’ throughout.
18. It would be very instructive to have max and min principle stress trajectories in both views. Von Mises is only the shear component and does not give a good indication of the loading patterns. Add supplementary figures with the principle stresses in both dorsal and ventral views.
19. Much of the discussion focuses on the FEA results, which are suspect because of boundary conditions. Reduce the emphasis on the magnitude of the stresses and focus more on the power of the combined methods. Very few groups use all of these technologies together. You appropriately emphasize that this is preliminary and outlines a workflow to improve upon.

---

## Round 0.2 · Minor Revisions

Thank for you for your extensive revisions on this manuscript. It is almost ready for acceptance. One of the reviewers has suggested a couple of very minor changes. One regards use of the term 'inverse dynamics'. I think the reviewer is correct in stating that you are not technically doing inverse dynamics. They suggest an alternative phrasing. However, I do not agree with the second point made by the reviewer with regard to the loading conditions. I think you have already addressed this point in the revisions and the sensitivity analysis that you included.

I don't expect it will take you long to make these final small changes, so I look forward to seeing your updated manuscript in the very near future.

Reviewer 3 ·

Basic reporting

Sufficient

Experimental design

"inverse dynamics" should be renamed 'Static analysis of joint reactions at mid-stance'. It is not inverse dynamics if you are assume m=0 and inertia = 0.

Many of the methodological questions have now been answered.

Validity of the findings

The authors findings are inherently based upon the loading conditions. They state that it is not possible to load so that it is fixed at the centroid. It is unclear why not. As it is modeled (cantilever beam), there will be a reaction moment (as demonstrated particularly in the short stubby beam).

---

## Round 0.3 · accepted · Accept

Thanks for your detailed comments on the use of the term 'inverse dynamics' and the addition of text to the manuscript to help explain your methods. I am now satisfied that your manuscript is ready to be accepted for publication. Congratulations on a great paper!